# Ischaemic preconditioning and pharmacological preconditioning with dexmedetomidine in an equine model of small intestinal ischaemia-reperfusion

Kathrin S. König[1]☯, Nicole Verhaar[1]☯, Klaus Hopster[1]☯¤, Christiane Pfarrer[2], Stephan Neudeck[1], Karl Rohn[3], Sabine B. R. Kästner[1,4]*

**1** Clinic for Horses, University of Veterinary Medicine Hannover, Hannover, Germany, **2** Institute for Anatomy, University of Veterinary Medicine Hannover, Hannover, Germany, **3** Department of Biometry, Epidemiology and Information Processing, University of Veterinary Medicine Hannover, Hannover, Germany, **4** Clinic for Small Animals, University of Veterinary Medicine Hannover, Hannover, Germany

☯ These authors contributed equally to this work.
¤ Current address: Department of Clinical Studies, University of Pennsylvania, School of Veterinary Medicine, Philadelphia, Pennsylvania, United States of America
* sabine.kaestner@tiho-hannover.de

**Data Availability Statement:** All data files (tables on individual histology scores and immunhistology cell counts are available from the Mendeley

## Abstract

Small intestinal strangulation associated with ischaemia-reperfusion injury (IRI) is common in horses. In laboratory animals IRI can be ameliorated by ischaemic preconditioning (IPC) and pharmacological preconditioning (PPC) with dexmedetomidine. The aim of this study was to determine the effect of PPC with dexmedetomidine or IPC in an equine model of small intestinal ischaemia-reperfusion (IR). In a randomized controlled experimental trial, 15 horses were assigned to three groups: control (C), IPC, and PPC with dexmedetomidine (DEX). All horses were placed under general anaesthesia and 90% jejunal ischaemia was induced for 90 minutes, followed 30 minutes of reperfusion. In group IPC, three short bouts of ischaemia and reperfusion were implemented, and group DEX received a continuous rate infusion of dexmedetomidine prior to the main ischaemia. Jejunal biopsies were collected before ischaemia (P), and at the end of ischaemia (I) and reperfusion (R). Mucosal injury was assessed by the Chiu-Score, inflammatory cells were stained by cytosolic calprotectin. The degree of apoptosis and cell necrosis was assessed by cleaved-caspase-3 and TUNEL. Parametric data were analyzed by two-way ANOVA for repeated measurements followed by Dunnetts t-test. Non parametric data were compared between groups at the different time points by a Kruskal-Wallis-Test and a Wilcoxon-2-Sample-test. The mucosal injury score increased during I in all groups. After reperfusion, IRI further progressed in group C, but not in IPC and DEX. In all groups the number of cleaved caspase-3 and TUNEL positive cells increased from P to I. The number of TUNEL positive cells were lower in group DEX compared to group C after I and R. Infiltration with calprotectin positive cells was less pronounced in group DEX compared to group C, whereas in group IPC more calprotectin positive cells were seen. In conclusion, IPC and DEX exert protective effects in experimental small intestinal ischaemia in horses.

database (https://data.mendeley.com/datasets/yjvj4vpmgc/1); DOI: 10.17632/yjvj4vpmgc.1.

**Funding:** This work was supported by Deutsche Forschungsgemeinschaft. The funders had no role in study design, data collection and analysis, decision to publish, or preparation of the manuscript.

**Competing interests:** The authors have declared that no competing interests exist.

## Introduction

Colic is one of the most common causes of death in horses [1,2]. Even though the management of these patients has improved in the last decades, small intestinal strangulations with concurrent ischaemia/reperfusion injury (IRI), are still associated with a relatively high mortality rate after surgery [3]. In human medicine, the concept of preconditioning has been instituted in the treatment and prevention of IRI in different tissues [4,5]. Ischaemic preconditioning (IPC) is defined as the activation of intrinsic cell survival programs after exposure to mild or brief ischemic stimuli [6]. The concept of IPC was first reported in 1986, where it was shown that brief periods of ischaemia reduced the myocardium infarct size in dogs to 25% of that seen in the control group [7]. The mode of action has been investigated, and an early and a late phase of protection have been identified. The first involves the activation of pre-existing effector molecules like prosurvival kinases Akt and ERK-1/2 [8]. The latter emerges after 12–24 hours and relies on the expression of distal mediators such as iNOS, HSP and COX-2 [9]. Many authors have reported beneficial effects of preconditioning in different experimental models for IRI, including the small intestine [10,11]. Until now, the implementation of IPC has not been documented in horses.

It has been found that preconditioning with different pharmacological agents can also exhibit a protective effect, currently known as pharmacological preconditioning (PPC). The alpha-2-agonist dexmedetomidine has been shown to ameliorate IRI in liver [12,13], myocardium [14], skeletal muscle [15] and small intestine [16] in laboratory animals. Different alpha-2-agonists including dexmedetomidine are used for sedation and partial intravenous anaesthesia as well as total intravenous anaesthesia in horses [17]. To the author´s knowledge, there are no reports on intestinal PPC with dexmedetomidine in horses.

Therefore, the aim of this study was to determine the effect of PPC with dexmedetomidine or IPC in an equine model of small intestinal ischaemia-reperfusion (IR). We hypothesized that IPC and PPC would ameliorate IRI, reflected by reduced morphological tissue injury and attenuated apoptosis and inflammation.

## Materials and methods

### Horses

Fifteen adult horses of various breeds (13 Warmbloods, 1 Trotter, 1 Quarter Horse) with a mean (SD) age of 6.2 (4.2) years and a mean body weight of 542 (62) kg were used for this study. Horses were adapted to the holding facilities for a minimum of 2 weeks prior to the study. They were fed 1.5–2% hay per kg body weight, and they had free access to water. Horses were determined to be free of gastrointestinal or other systemic disease by physical examination and fecal egg count. The majority of the horses were scheduled for euthanasia because of chronic orthopedic disease. Food, but not water was withheld 4 hours prior to anaesthesia. After completing the experiments, the horses were euthanized by 80 mg/kg BW pentobarbital (Euthadorm®, CP-Pharma pharmaceuticals company, Burgdorf, Germany) intravenously while still being anaesthetized. The horses´s bodies were used for teaching in anatomy classes. The experimental protocol was reviewed and approved by the Ethics Committee for Animal Experiments of Lower Saxony, Germany according to national and EU animal protection regulations (No. 33.12-42502-04-15/1834).

### Study design

The study was carried out as a randomized controlled experimental trial. The horses were randomly divided into three groups of five horses each. The control group (group C) received no

additional treatment. In group IPC, the intestine was preconditioned with brief episodes of ischaemia prior to the main ischaemic event. The horses in group DEX were pharmacologically preconditioned and received a continuous-rate-infusion (CRI) with dexmedetomidine.

## Anaesthesia

A 12-gauge catheter (Intraflon 2, 12G-80mm, VYGON GmbH & Co.KG, France) was placed aseptically in one of the jugular veins. All horses were premedicated with guaifenesine (80–100 mg/kg i.v., My-50 mg/ml, CP-Pharma pharmaceuticals company, Burgdorf, Germany) until they became ataxic. Subsequently, general anaesthesia was induced with ketamine (2.2 mg/kg BWT IV, Narketan®, Vétoquinol GmbH, Ravensburg, Germany) and midazolam (0.05 mg/ kg BWT IV, Midazolam®, B. Braun, Melsungen AG, Melsungen, Germany). Horses were endotracheally intubated and general anaesthesia was maintained with isoflurane (Isofluran®, CP-Pharma pharmaceuticals company, Burgdorf, Germany) in 100% oxygen with an end-expiratory isoflurane concentration of 1.3–1.6 vol % to achieve an adequate depth of anaesthesia. In group DEX, the horses received a continuous-rate-infusion (CRI) with dexmedetomidine at a rate of 0.007 mg/kg/ h (Dexdomitor®, 0.5 mg/ml, Orion Pharma, Finland), initiated immediately after induction of anaesthesia. All horses were ventilated mechanically (positive inspiratory pressure of 20–25 cmH$_2$O) aiming for normocapnia (PaCO$_2$ between 35 and 45 mmHg). Lactated Ringer´s solution (Ringer-Laktat®, B. Braun Melsungen AG, Melsungen, Germany) at a rate of 5 ml/kg/h—15 ml/kg/h and dobutamine (Dobutamin-ratiopharm 250 mg®, Ratiopharm pharmaceutical company, Ulm, Germany) at a rate of 0.5–1.0 µg/kg/h were administered to maintain the mean arterial blood pressure above 60 mmHg.

## Study protocol and sample collection

After induction of general anaesthesia, the horses were placed in dorsal recumbency. After an equilibration period of 60 minutes, a ventral midline laparotomy was performed and the abdomen and small intestines were checked for abnormalities. At 90 minutes after induction, a 20-cm intestinal segment was resected from the middle part of the jejunum (pre-ischaemia sample P). In between each stage of the procedure, the intestines were placed back into the abdomen and the incision was transiently closed with towel clamps.

Subsequently, ischaemia was induced in approximately one meter of terminal jejunum by occluding the intestinal loops and associated mesentery with umbilical tape (Scheidenband nach Bühner, WDT, Garbsen, Germany). The umbilical tape was tightened until the blood flow was reduced to 10% of the baseline value. During ligation, the blood flow and oxygen saturation in the intestinal tissue were measured by laser-doppler flowmetry and micro-lightguide spectro-photometry (Oxygen 2 See, LEA Medizintechnik GmbH, Gießen, Germany) as described previously [18]. In group IPC, ischaemic preconditioning was performed before initiating the main ischaemic event, by implementing three cycles of ischaemia alternated with three cycles of reperfusion of 2 minutes each. In all groups, ischaemia was maintained for 90 minutes, after which the umbilical tape was cut and reperfusion followed for 30 minutes. An intestinal segment was resected from the ischaemic section just before the ligature was cut (ischaemia sample I), and at the end of reperfusion (reperfusion sample R). Immediately after resection, the intestinal samples were flushed with warm isotonic saline solution, and full thickness intestinal biopsies were taken from the mesenteric and anti-mesenteric part of the intestine.

## Histopathology

Tissue samples were fixed in neutral buffered formalin for 48 hours and subsequently embedded in paraffin. After routine processing, 3–4 µm thick sections were cut and mounted onto

slides. The slides were stained routinely with hematoxylin and eosin (HE). Mucosal injury was scored by a blinded, investigator trained and supervised by board certified pathologist according to a modified Chiu scoring system [19,20]. The grades of this score are defined as follows: grade 0: normal mucosa histomorphology; grade 1: slight separation of epithelial cells from the lamina propria at the tip of the villus. A subepithelial space has formed at the tip of the villus ("Gruenhagen's space"); grade 2: extension of the subepithelial space and minimal demarcation between epithelial cells and lamina propria with partial loss of the epithelial cells at the tip of the villus; grade 3: the epithelial separation from the lamina propria has progressed from the tip towards to the base, exposing one-third to one-half of the lamina propria, moderate vasodilation and congested capillaries in lamina propria and tela submucosa; grade 4: nearly complete to complete loss of epithelium with marked vasodilation, congested capillaries und hemorrhage in the lamina propria and edema in the tela submucosa; grade 5: complete loss of villus architecture leaving an irregular denuded surface, disintegration of the lamina propria, necrosis of the crypts (Fig 1). For every horse, one slide per time point of both the mesenterial and anti-mesenterial biopsies was assessed using light microscopy at a 400-fold magnification (AXIO Scope. A1; Carl Zeiss Microscopy GmbH, Jena). Per slide, the mucosal injury was scored in duplicate (KK) in 5 adjoined high-power-fields (HPF's) and averaged for each slide.

## Immunohistochemistry

Apoptotic cells were detected by immune-histochemical staining for cleaved-caspase-3 with Cleaved (CL)-Caspase-3 (Asp175) (Cell Signaling Technology, Inc. Danvers, USA) as previously described (Fig 2A). After routine deparaffinizing and blocking the endogenous peroxidases, sections were washed in Tris Buffered Saline with Tween (TBS-T) three times for 5 minutes each. Then, specimens were immersed in citrate unmasking solution while heated in a microwave at 800W for 20 minutes. After cooling down to room temperature (RT), the sections were rinsed in TBS-T three times for 5 minutes each and covered with 1:5 normal goat serum in phosphate-buffered-saline (PBS) for 20 minutes. This was followed by the application of cleaved caspase-3 antibody diluted 1:200 in PBS with 1% Bovine Serum Albumin (BSA; Sigma Aldrich Chemie GmbH, Steinheim, Germany) and incubation overnight at 4°C. The negative control was incubated with PBS with 1% BSA. The sections were returned to RT and washed in PBS three times for 5 min each. Subsequently, 1:200 Goat-anti-rabbit antibody (Vector Laboratories, Burlingame, USA) was applied and the sections were incubated for 30 minutes. The specimens were washed in PBS three times for 5 minutes each, then covered with avidin-biotin complex (Biologo, Kronshagen, Germany) and incubated for 30 min. Specimens were washed in PBS two times for 5 minutes each and submersed in PBS solution. Slides were covered with diaminobenzidine for about 1 min to achieve color development under visual control by light microscopy. The specimens were counterstained with modified hematoxylin (Delafield Hemalaun).

Fragmented DNA was stained by the terminal deoxynucleotidyl transferase (TdT)-mediated dUTP-digoxigenin nick-end labeling (TUNEL)–method using the ApopTag® Peroxidase In Situ Apoptosis Detection Kit S7100 (Merck Millipore SAS, Molsheim, France) according to the manufacturer´s instructions (Fig 2B). The specimens were routinely dewaxed and washed in PBS for 5 minutes, prior to incubation with proteinase K (20 µl/ml) for 15 minutes. Then specimens were washed in distilled water twice for 2 minutes each before inhibiting endogenous peroxidase with 3.0% hydrogen peroxide in PBS for 5 minutes. Specimens were washed in PBS twice for 5 minutes each, then the Equilibration Buffer was applied directly on the specimens for 10 minutes. Followed by application of diluted TdT-Enzyme for one hour at 37°C. The negative control was incubated with Reaction Buffer without TdT- Enzyme. To

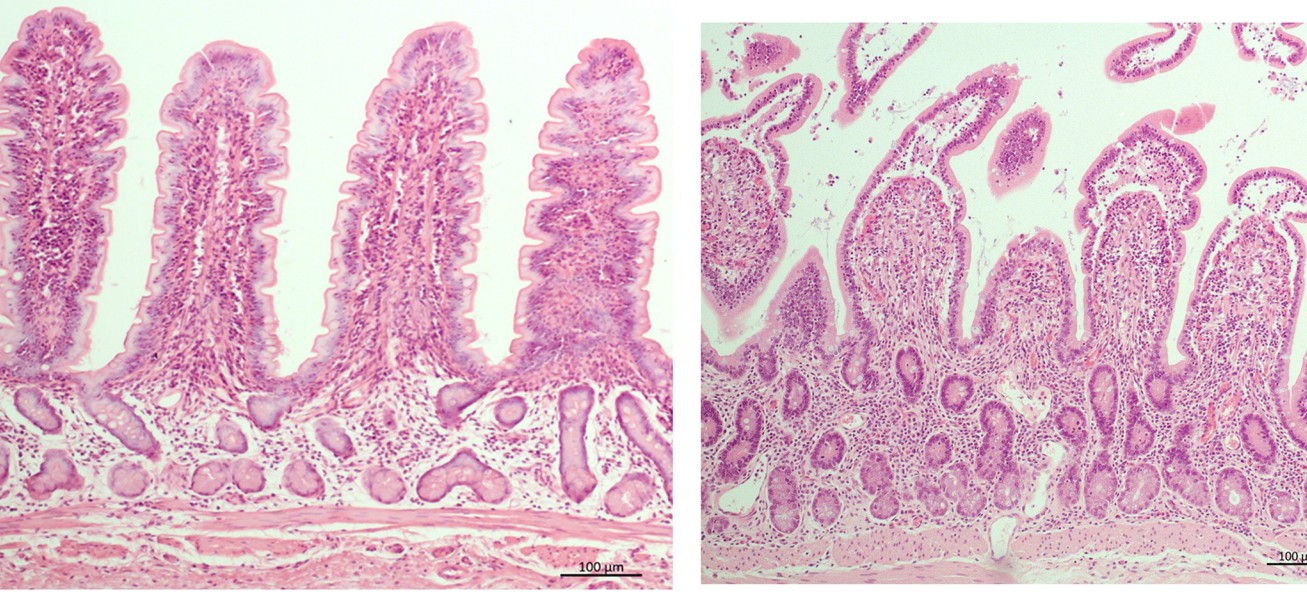

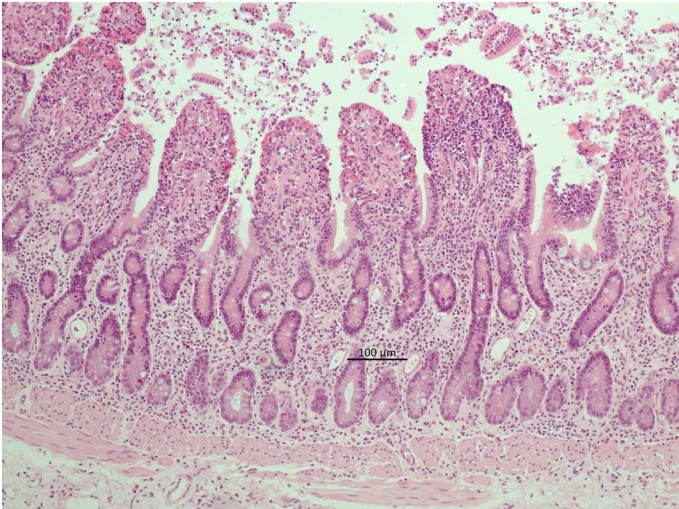

**Fig 1. Representative microscopic images of the intestinal mucosa stained with HE, characterized according to a modified Chiu score (Grade 0–5; 0 = normal; 5 = complete loss of villi).** (A) Pre-ischemia sample, grade 0; (B) Ischaemia sample with epithelial separation from the lamina propria over one-third to one-half of the lamina propria, grade 3; (C) Ischaemia sample, with near to complete epithelial separation and cellular debris in the intestinal lumen, grade 4.

determine the reaction, the sections were immersed in Stop/ Wash Buffer diluted 1:34 in distilled water for 10 minutes. The negative control was immersed separately. Then specimens were washed in PBS three times for one min each and covered with Anti-Digoxigenin-Peroxidase for 30 minutes, followed by washing specimens in PBS four times 2 min each. Sections were covered with diaminobenzidine for about 20 s under visual control by light microscopy to achieve color before they were counterstained with modified haematoxylin (Delafield Hemalaun).

The inflammatory cell count was assessed by immune-histochemical staining for cytosolic calprotectin using monoclonal mouse anti-human myeloid/histiocyte antigen (clone MAC 387) as previously described [21]. Briefly, the slides were immersed in descending ethanol series and dewaxed, before endogenous peroxidase was blocked by hydrogen peroxide in

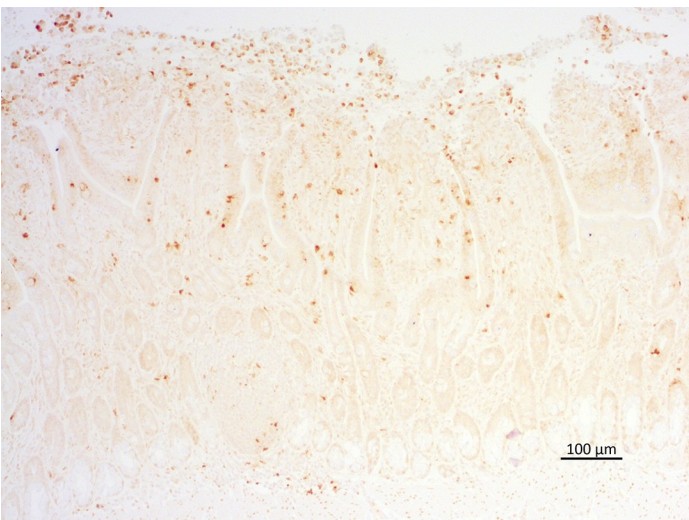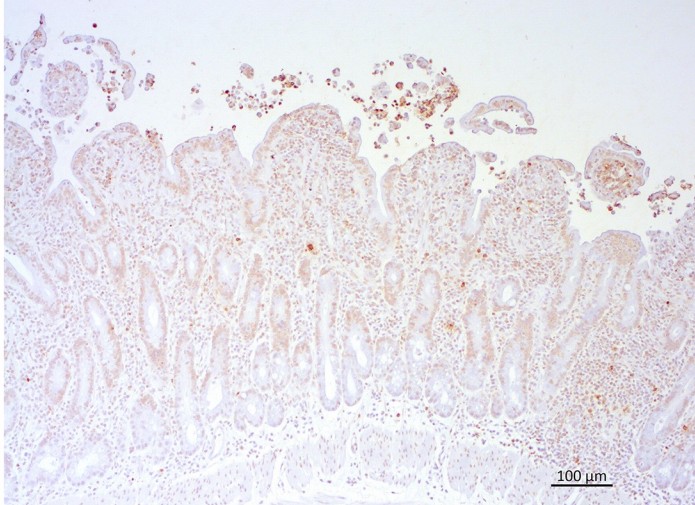

**Fig 2.** Representative microscopic images of the intestinal mucosa after ischaemia with cleaved caspase-3 staining (A) and TUNEL staining (B). The arrows indicate cleaved caspase 3 positive cells or TUNEL-positive cells.

ethanol (0.5%) for 30 minutes. Then, the sections were submersed in citrate buffer (pH 6) at 95°C, followed by cooling down to RT. Subsequently, the slides were covered with normal goat serum for 20 minutes, then primary antibody was applied and slides were incubated for 18 h at 4°C. Slides were covered with biotinylated secondary antibody (goat anti-mouse IgG from Vector Laboratories) for 30 minutes, followed by administering avidin-biotin complex (Vector Laboratories) for 30 minutes, according to the manufacturer´s specifications. Between each step, slides were washed in PBS three times. Finally, sections were covered with diamino-benzidine and hydrogen peroxide for 5 minutes to achieve color before they were counter-stained with modified hematoxylin (Delafield Hemalaun). Equine lymph node tissue served as positive control in all three immuno-histochemical staining protocols.

The examination of the stained slides was always performed by the same blinded investigator. The positive cells in a defined mucosal area of 5 to 10 adjoined HPFs per slide were counted. The exact surface area was measured by use of a microscope camera and accompanying software (Axiocam 105 color and Software ZEN 2.3; AXIO Scope.A1; Carl Zeiss Microscopy GmbH, Jena), and the cell count was expressed as cells/mm$^2$. For the calprotectin stain, an additional count of the positive cells in the serosa over the width of 5 HPF´s was performed. Furthermore, the positive cells in 30 adjoined submucosal venules were counted.

## Data handling/statistical analysis

Statistical analysis were performed by SAS 9.4 (SAS Institute Inc. Cary, USA) and GraphPad PRISM 8.01 (GraphPad Software, Inc. USA). A p-value $< 0.05$ was regarded as statistically significant. Data distribution was assessed by visual evaluation of qq-plots of model residuals and the Kolmogorov-Smirnov test. Data are presented as mean and standard deviation (SD) or median [minimum–maximum].

Normal distributed results of the mesenteric and anti-mesenteric small intestinal samples were compared by t-test for paired observations (cell counts) and Wilcoxon matched-pairs signed rank test (histopathology score). No differences could be detected, therefore, samples were pooled for further analysis. To correct for the variation in preexisting apoptotic and inflammatory cells, relative cell counts were calculated. The cell count of the pre-ischaemia sample was set to 1 in each individual horse and relative changes were calculated.

The endtidal isoflurane concentration was averaged over time (sum of measurements/number of measurements) and compared between the groups by one-way-ANOVA. The number of calprotectin positive cells were log transformed to achieve normality.

Parametric data were analyzed by two-way ANOVA for repeated measurements followed by Dunnetts t-test where appropriate. Non parametric data (histology score) were compared between the groups at the different time points by a Kruskal-Wallis-Test and a Wilcoxon-2-Sample-test (for multiple paired comparison). To assess the differences between the time points within groups, a (Friedmans-) Permutation-test was used, and the p-values were adjusted according to Sidak.

## Results

Cardiovascular and respiratory parameters showed minor variation within the accepted normal range during anaesthesia. The end-expiratory isoflurane concentrations to maintain an adequate depth of anaesthesia were 1.5 (0.05) vol %, 1.5 (0.1) vol % and 1.4 (0.1) vol % in group C, IPC and DEX, respectively, with a significant difference between IPC and DEX ($p = 0.04$).

### Histopathology

Mucosal injury increased from baseline to ischaemia in group C ($p = 0.004$), IPC ($p = 0.032$) and DEX ($p = 0.040$) (Fig 3). After reperfusion, IRI further progressed in group C ($p = 0.010$). In group IPC there was no difference between ischemia and reperfusion, whereas in group DEX a significant decrease in IRI from ischemia to reperfusion occurred. In the pre-ischaemia and reperfusion sample, the histopathology score did not differ between the groups. After ischaemia, group DEX had a significantly higher score compared to group C ($p = 0.027$). One horse in group DEX and one in group IPC showed a score of 0 after reperfusion.

### Immunohistochemistry

In all pre-ischaemia samples, cleaved caspase-3 and TUNEL positive cells were detected. In all treatment groups the number of caspase positive cells increased 2–3 fold from pre-ischaemia

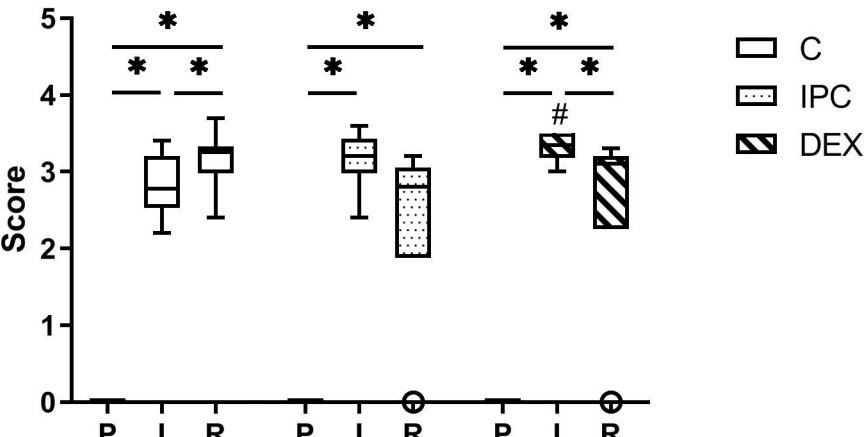

**Fig 3. Tukey box and whisker plot of the effect of dexmedetomidine and ischaemic preconditioning on mucosal injury score (modified Chiu score, grade 0–5) in an equine intestinal IR model.** * indicates a significant difference within groups ($p < 0.05$), # indicates a significant difference compared to the control group at the same time point ($p < 0.05$); P = pre-ischaemia; I = ischaemia; R = reperfusion. Boxes represent the interquartile range, the horizontal line is the median, the whiskers indicate 1.5 x the interquartile range, open circles represent outliers.

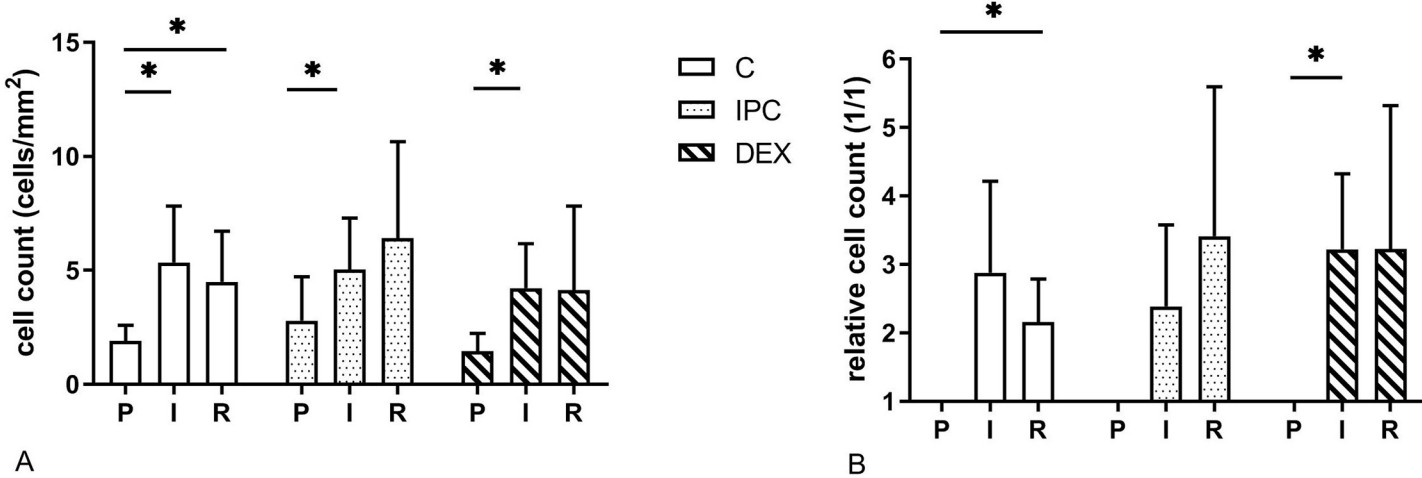

**Fig 4.** Mean (SD) of the effect of dexmedetomidine and ischaemic preconditioning on absolute (A) and relative (B) number of cleaved caspase 3 positive cells in an equine intestinal IR model. * indicates a significant difference to the pre-ischemia sample within groups (p < 0.05), P = pre-ischaemia; I = ischaemia; R = reperfusion.

to ischaemia (p = 0.046, p = 0.02 and p = 0.019 in group C, IPC and DEX, respectively) (Fig 4). In group C, the number of caspase positive cells after reperfusion differed from the pre-ischaemia sample (p = 0.039). The number of caspase positive cells were not different among the treatment groups at any time point.

The number of TUNEL stained cells were significantly increased after ischaemia in all groups (p = 0.027, p = 0.027 and p = 0.043 in group C, IPC and DEX, respectively) (Fig 5). There was an overall treatment effect (p = 0.007). The number of TUNEL positive cells were lower in group DEX compared to group C after ischaemia (p = 0.036) and after reperfusion (p = 0.021).

There were calprotectin positive cells in the mucosa of all samples (Table 1). In the ischaemia sample, the relative cell count was significantly lower in group IPC compared to group C (p = 0.047). In group IPC and DEX, the mucosal cell count increased significantly after reperfusion, compared to the pre-ischaemia sample (p = 0.045 and p = 0.038, respectively). In both

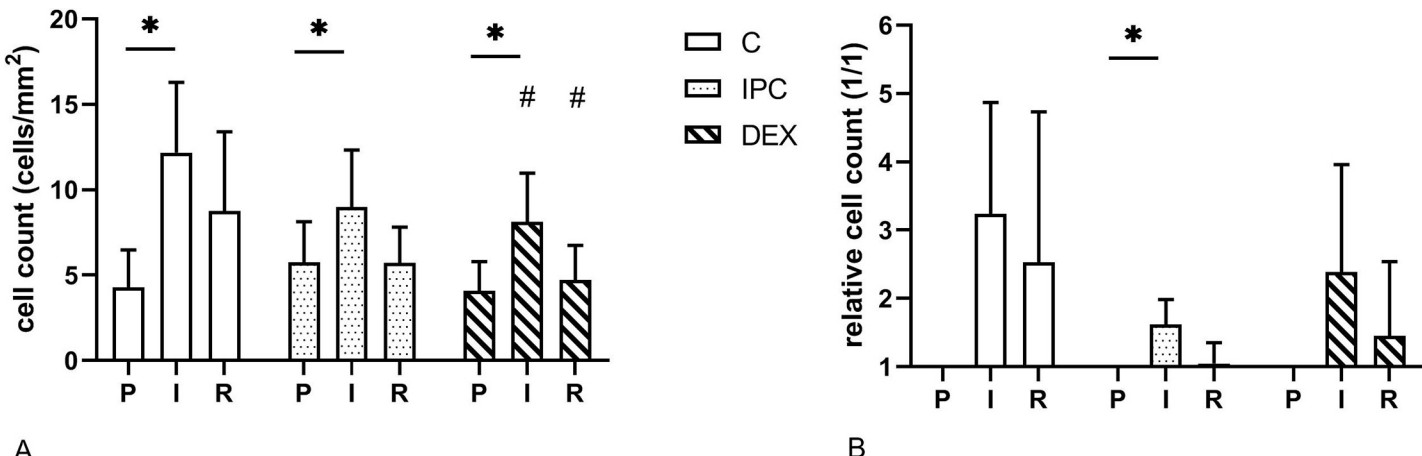

**Fig 5.** Mean (SD) of the effect of dexmedetomidine and ischaemic preconditioning on absolute (A) and relative (B) number of TUNEL positive cells in an equine intestinal IR model. * indicates a significant difference to the pre-ischemia sample within groups (p < 0.05), # indicates a significant difference compared to the control group at the same time point (p < 0.05); P = pre-ischaemia; I = ischaemia; R = reperfusion.

**Table 1. Median and range of calprotectin positive cells in small intestinal mucosa, submucosa and serosa in 15 horses before (P) and after ischemia (I) and after reperfusion (R).**

| | Mucosa | | |
|---|---|---|---|
| | Group C | Group IPC | Group DEX |
| Pre-ischaemia (cells/mm$^2$) | 1 [0.4–9.9] | 16.3 [6.2–22.3] # | 12.5 [0.5–31.7] |
| Relative (1/1) | 1 | 1 | 1 |
| Ischaemia (cells/mm$^2$) | 6.7 [5.8–19.2] | 30.2 [13.9–124.0] | 25.6 [5.1–37.8] |
| Relative (1/1) | 6.6 [1.9–22.2] | 3.7 [0.8–5.6] # | 2.1 [1–11.8] |
| Reperfusion (cells/mm$^2$) | 13.1 [4.5–29.7] | 67.5 [25.0–107.7]* # | 33.5 [6.2–89.6]* |
| Relative (1/1) | 12.8 [0.4–74.2] | 4.1 [4.0–6.6] | 2.8 [1.9–25.1] |
| | Submucosa | | |
| | Group C | Group IPC | Group DEX |
| Pre-ischaemia (cells/mm$^2$) | 1 [1–10] | 1 [1–1] | 0 [0–1] |
| Relative (1/1) | 1 | 1 | 1 |
| Ischaemia (cells/mm$^2$) | 5 [2–11] | 7 [5–11]* | 1 [0–3] |
| Relative (1/1) | 1.5 [1–4] | 4 [3–6]* | 1 [1–3] |
| Reperfusion (cells/mm$^2$) | 14 [7–41] | 23 [6–30]* | 4 [2–9] # |
| Relative (1/1) | 4 [1.4–21] | 12 [3.5–15.5]* # | 3 [2.5–10] |
| | Serosa | | |
| | Group C | Group IPC | Group DEX |
| Pre-ischaemia (cells/mm$^2$) | 0 [0–4] | 1 [0–1] | 2 [0–5] |
| Relative (1/1) | 1 | 1 | 1 |
| Ischaemia (cells/mm$^2$) | 1 [0–4] | 12 [7–34] # | 2 [0–8] |
| Relative (1/1) | 2 [0.2–5] | 7.5 [4–35] # | 1 [2–6] |
| Reperfusion (cells/mm$^2$) | 87 [45–125]* | 123 [66–503]* | 116 [22–169]* |
| Relative (1/1) | 47 [19.8–126]* | 62 [25–151]* | 23 [13.6–170]* |

Number of calprotectin positive cells (median, range) in the intestinal mucosa (per mm$^2$), submucosa (30 venules) and serosa (over 5 HPFs) at three different time-points, expressed as absolute number of cells and relative change (1/1) compared to the pre-ischaemia sample.

*indicates a significant difference to pre-ischaemia within the same group (p< 0.05)

# indicates a significant difference compared to the control group at the same time point (p < 0.05).

the submucosal venules and the serosa, group IPC had a significantly higher relative cell count after ischaemia, compared to group C (p = 0.032 and 0.016, respectively) and group DEX (p = 0.016 for both locations) (Table 1). In the submucosa, only group IPC showed a significant increase in cell count during ischaemia (p = 0.026), and there was no further increase during reperfusion. In the serosa, the inflammatory cell count increased significantly over time in all groups (p<0.05). After reperfusion, there were no differences between the groups.

## Discussion

This is the first study describing intestinal IPC and PPC with dexmedetomidine in horses. We hypothesized that these treatments would ameliorate IRI reflected by histomorphology and less apoptosis initiation and invasion of inflammatory cells. Both IPC and PPC prevented the progression of morphological tissue injury from ischaemia to reperfusion, whereas the presence of late stage apoptotic and necrotic cells and the invasion of inflammatory cells was reduced after PPC with dexmedetomidine, but not after IPC. Overall these observations indicate a protective effect of IPC and PPC, confirming the hypothesis that these treatments can reduce IRI in the equine small intestine.

The histomorphology score according to Chiu [22] qualitatively assesses the progressive separation of epithelial cells from the villi, denudation of the villi and finally destruction of the underlying lamina propria. Denudation of villi (score 4) occurred rarely and destruction of the lamina propria (score 5) were not seen in our model indicating that 90 minutes of incomplete intestinal ischaemia with 90% flow reduction without intestinal distension induced a moderate injury. Only in a moderate grade injury model without massive tissue necrosis a protective effect of IPC or PPC could elicit protective effects. Effects on re-epithelization processes and tissue healing could not be assessed with our study design because of the short observation period after reperfusion.

In the untreated control group, the mucosal histomorphology score increased during reperfusion, whereas group IPC and DEX did not show a progression in mucosal injury during reperfusion. In group DEX even a significant decrease in histomorphology score occurred during reperfusion, but the latter needs to be interpreted with caution, as the DEX group started with a higher score after ischemia compared to the control group.

Numerous studies in laboratory animals have shown the protective effects of IPC [7,23,24]. The current study identified a possible protective effect of the method in intestinal tissue of horses. Various studies have been done in laboratory animals on the timing and duration of the ischaemic bouts. Varga et al. [25] compared two different ischaemic preconditioning regimes and concluded that fewer and longer periods of IPC with 2 ischaemic cycles of 12 minutes each resulted in less severe mucosal injury compared with 4 cycles of 4 minutes ischaemia. On the other hand, Kageyama et al. [26] employed 2 cycles of 4 minutes of ischaemia with protective effects. Furthermore, it has been shown that repeated episodes of ischaemia induce a better protective effect than a single ischaemic period [27]. One study used the same IPC protocol in rats as employed in the current study, with significant improvement of tissue injury in the IPC group compared with the IR group [28]. Overall, the different studies investigating IPC are not directly comparable because diverse IPC protocols and IR models were used. Moreover, the precise mechanism of IP remains obscure, and it cannot be excluded that there are differences between organs and species.

Preconditioning effects have been described for several pharmacologic agents including dexmedetomidine [12,29,30]. A reduction in mucosal injury after reperfusion could also be seen in the current study. The exact mechanism of the protective effect of dexmedetomidine is not completely clear. Both the alpha-2-receptor and the imidazolin-1-receptor could convey protection [31]. Okada et al. [32] believe that preconditioning might be triggered by an ischaemic effect mediated by alpha-2 adrenergic stimulation and consecutive vasoconstriction making it similar to IPC. The activation of hypoxia-inducible factor 1-alpha might play a role; however, this is still under debate [33,34]. Dexmedetomidine has been shown to inhibit lipid peroxidation of the cell membranes in different tissues [35,36], and one study reports that dexmedetomidine has anti-apoptotic and anti-inflammatory effects in intestinal tissue following bowel injury. In the current study, a 2 to 3-fold increase in cleaved caspase 3 stained cells as an indicator of early apoptosis stages occurred after ischaemia and reperfusion in all groups in a similar way, however, the number of late apoptotic or dead cells indicated by TUNEL stain were reduced in the dexmedetomidine treated horses. The infiltration of calprotectin positive inflammatory cells, which comprise mainly neutrophils, but also macrophages and monocytes [37,38] was also lower in group DEX. These observations corroborate the anti-inflammatory and anti-apoptotic effects of dexmedetomidine in equine intestinal tissue as well. Dexmedetomidine preconditioning effects might be dose dependent, but in the current study only 1 dose was investigated. So far the majority of studies investigating preconditioning with dexmedetomidine are performed in laboratory animals, where the alpha-2 agonist is given as a single intraperitoneal injection, which leads to a different pharmacokinetic profile as an intravenous

infusion and might also induce local intestinal effects [30,36]. The dose used in the current study was based on doses used for balanced anaesthesia in horses but without a loading dose to ameliorate the cardiovascular effects [17,39]. Plasma or tissue concentrations of dexmedetomidine were not determined in the current study, but an equilibration time of 90 minutes was allowed from start of the dexmedetomidine infusion until induction of ischaemia, which approximates 5x the elimination half-life of dexmedetomidine [40,41], so that steady state dexmedetomidine concentrations could be expected.

Volatile anaesthetics including isoflurane have also been shown to exhibit dose dependent preconditioning effects [42]. Therefore, the degree of mucosal injury and apoptosis might be blunted across all groups, making the detection of a treatment effect more difficult. The mean isoflurane concentration required to maintain adequate anesthetic depth was about 0.1 vol% lower in group DEX compared to the other groups, explained by MAC-reduction effects of alpha-2-agonist balanced anaesthesia [43]. It is not clear if this difference in endtidal isoflurane concentration could have a significant effect on the degree of isoflurane induced PPC effects.

The number of inflammatory cells as indicated by calprotectin positive cells in the different intestinal tissue layers was higher in the IPC treated horses, which might be a sequela of more or repeated intestinal tissue handling in this group [44]. The clinical relevance of this observation remains unclear.

Both cleaved-caspase 3 and TUNEL stain can indicate apoptotic processes, whereby caspase 3 represents an earlier stage of apoptosis than TUNEL. In addition, the TUNEL stained cells may not only represent apoptotic cells, as DNA fragmentation also occurs during cell necrosis [45]. For the assessment of the tissue sections, only cells that were still attached to the mucosa were counted. The cell debris in the intestinal lumen was not included in the cell count, which could have influenced the results in case of severe villous destruction and epithelial loss, leading to an underestimation of apoptotic/necrotic cells.

A limitation of this study is the small sample size. Even though a power analysis based on expected difference in the Chiu score was performed prior to executing the study, there may still be a statistical type II error for the number of apoptotic or inflammatory cells, failing to detect differences between the groups. Given the observed effect size and an alpha of 5% for the number of cleaved caspase-3 positive cells in the intestinal tissue, a power of 61% was achieved, to obtain 80% power 7 horses per group would have been required.

Another limitation of the current study was the short reperfusion time. Therefore, only short term changes could be observed, and a possible late phase of protection could not be assessed. The duration of experimental IRI depends on the model used. The current experimental model was effective in creating significant ischaemic injury in 90 minutes of 90% occlusion, reflected as a significant rise in histomorphology score. Previous studies looking at experimental small intestinal IRI in horses, employed models of complete ischaemia [20], low-flow ischaemia, reducing the blood flow by 75–80% [46] or the degree of ischaemia is not specified. Therefore, a direct comparison across studies is difficult. Ninety minutes of ischaemia resulted in a significant elevation of inflammatory cell counts [47] and calprotectin positive cells peaked within submucosal venules in equine large colon after a reperfusion period of 30 minutes [38].

In 1 horse in group DEX and in 1 horse in group IPC, the Chiu-Score was found to be grade 0 after reperfusion indicating complete recovery, even though there had been significant injury in the ischaemia sample. A mix-up of tissue samples was ruled out, therefore the samples were not excluded from analysis. It is known that first repair processes of epithelial restitution begin within minutes after injury [48], and villus remnants can partly or completely be lined with regenerating epithelium 12 hours after strangulating obstruction in ponies [49]. However, complete morphologic regeneration of the mucosa would be unlikely after complete

villus denudation after the limited reperfusion time. Possible explanations for the unaltered villus structure could be a combined effect of an incomplete loss of epithelium and the preconditioning effect, a disparity in the local blood flow, or a variation in the response to ischaemia of different intestinal segments.

## Conclusions

This study has shown that protective effects can be elicited by IPC and PPC with dexmedetomidine in experimental small intestinal ischaemia in horses. Preconditioning could be used in the management of equine colic, for example in the time span from diagnosis to surgical reduction of the lesion. The follow up time was limited in this study and in a clinical situation pre-emptive use of this concept is limited. Therefore, expansion to post-conditioning after onset of the ischaemia might have more clinical relevance and should be part of further investigations.

## Acknowledgments

The authors would like to thank Dr. A. Rötting, PhD, for excellent surgical support. We also thank Prof. Dr. A. Beineke (Institute of Pathology, University of Veterinary Medicine Hannover) for help with setting up the immunohistochemical techniques and D. Voigtländer and M. Langeheine (Institute for Anatomy, University of Veterinary Medicine Hannover, Hannover, Germany) for technical support.

This manuscript represents a portion of a thesis submitted by Dr. Kathrin König to University of Veterinary Medicine Hannover, Foundation, Hannover, Germany, as partial fulfillment of the requirements for a *Doctor medicinae veterinariae* degree.

Presented as abstract at the Association of Veterinary Anaesthetists Autumnn Meeting, Berlin, Germany, November 2017 and as poster at the German Veterinary Association Innlab Meeting, Hannover, Germany, February 2018.

## Author Contributions

**Conceptualization:** Klaus Hopster, Sabine B. R. Kästner.

**Data curation:** Nicole Verhaar, Sabine B. R. Kästner.

**Formal analysis:** Kathrin S. König, Karl Rohn, Sabine B. R. Kästner.

**Investigation:** Kathrin S. König, Klaus Hopster, Stephan Neudeck.

**Methodology:** Kathrin S. König, Christiane Pfarrer, Sabine B. R. Kästner.

**Resources:** Sabine B. R. Kästner.

**Software:** Karl Rohn.

**Supervision:** Christiane Pfarrer, Sabine B. R. Kästner.

**Visualization:** Kathrin S. König.

**Writing – original draft:** Kathrin S. König, Nicole Verhaar.

**Writing – review & editing:** Nicole Verhaar, Sabine B. R. Kästner.

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
