## [Decision Letter · Decision Letter 0]

16 Mar 2020

PONE-D-19-29159

Ischaemic preconditioning and pharmacological preconditioning with dexmedetomidine in an equine model of small intestinal ischaemia-reperfusion

PLOS ONE

Dear Dr Sabine Kästner,

Thank you for submitting your manuscript to PLOS ONE. After careful consideration, we feel that it has merit but does not fully meet PLOS ONE’s publication criteria as it currently stands. Therefore, we invite you to submit a revised version of the manuscript that addresses the points raised during the review process.

Dear Authors sorry for the delay in the review process but it was not easy to find the right reviewers. At the end the two reviewers were very satisfied by your manuscript, and only one suggested some minor revisions. Please try to respond to all the requests corrections in the pdf file

We would appreciate receiving your revised manuscript by Apr 30 2020 11:59PM. To enhance the reproducibility of your results, we recommend that if applicable you deposit your laboratory protocols in protocols.io, where a protocol can be assigned its own identifier (DOI) such that it can be cited independently in the future. For instructions see: http://journals.plos.org/plosone/s/submission-guidelines#loc-laboratory-protocols

We look forward to receiving your revised manuscript.

Kind regards,

Francesco Staffieri

Academic Editor

PLOS ONE

Journal Requirements:

2. In your Methods section, please provide additional details regarding the animals used in your study and ensure you have described the source. For more information regarding PLOS' policy on materials sharing and reporting, see https://journals.plos.org/plosone/s/materials-and-software-sharing#loc-sharing-materials.

3.  Thank you for including your ethics statement:

"The experimental protocol was reviewed by the Ethics Committee for Animal Experiments of Lower Saxony, Germany and approved according to national and EU animal protection regulations (No. 33.12 42502 04 15/1834).

This was a terminal study. Horses were euthanized being under general anesthesia by i.v. injection of 80 mg/kg BW sodium pentobarbital "

a) Please amend your current ethics statement to confirm that your named ethics committee specifically approved this study.

For additional information about PLOS ONE submissions requirements for ethics oversight of animal work, please refer to http://journals.plos.org/plosone/s/submission-guidelines#loc-animal-research  

4. We noted in your submission details that a portion of your manuscript may have been presented or published elsewhere.

 " This manuscript represents a portion of a thesis submitted by Dr. Kathrin König to University of Veterinary Medicine Hannover, Foundation, Hannover, Germany, as partial fulfillment of the requirements for a Doctor medicinae veterinariae degree. Thesis in German

Presented as abstract at the Association of Veterinary Anaesthetists Autumnn Meeting, Berlin, Germany, November 2017 and as poster at the German Veterinary Association Meeting, specialist group for Internal medicine and clinical laboratory diagnostics, Hannover, Germany, February 2018. Abstracts ".

"This publication was supported by Deutsche Forschungsgemeinschaft (DFG) and the

453 University of Veterinary Medicine Hannover, Foundation within the funding programme Open

454 Access Publishing.".

i) We note that you have provided funding information that is not currently declared in your Funding Statement. However, funding information should not appear in the Acknowledgments section or other areas of your manuscript. We will only publish funding information present in the Funding Statement section of the online submission form.

ii) Please remove any funding-related text from the manuscript and let us know how you would like to update your Funding Statement. Currently, your Funding Statement reads as follows:

"The authors received no specific funding for this work.".

Additional Editor Comments (if provided):

Dear Authors

sorry for the delay in the review process but it was not easy to find the right reviewers.

At the end the two reviewers were very satisfied by your manuscript, and only one suggested some minor revisions. Please try to respond to all the requests corrections in the pdf file

Reviewers' comments:

Reviewer's Responses to Questions

**Comments to the Author**

1. Is the manuscript technically sound, and do the data support the conclusions?

Reviewer #1: Yes

Reviewer #2: Yes

2. Has the statistical analysis been performed appropriately and rigorously? 

Reviewer #1: Yes

Reviewer #2: Yes

3. Have the authors made all data underlying the findings in their manuscript fully available?

Reviewer #1: Yes

Reviewer #2: Yes

4. Is the manuscript presented in an intelligible fashion and written in standard English?

Reviewer #1: Yes

Reviewer #2: Yes

5. Review Comments to the Author

Reviewer #1: The authors have to be applauded for a very good manuscript. It is clear and concise and right to the point. It deals with a very important aspect of equine surgery and medicine and therefore, mertits publication.

All chapters are written in a good understandable English, and are also well structured. Methods are described very clearly and are current with modern literature. A thorough literature search has also been performed and results critically discussed with results of other studies.

Conclusions are correct and according to their real findings.

Only a few additions or corrections are recommended (see comments in manuscript).

The manuscript can be accepted without a further review again.

Reviewer #2: The aim of this study was to determine the effect of PPC with dexmedetomidine or IPC in an equine model of small intestinal ischaemia-reperfusion. This study has shown that protective effects can be elicited by IPC and PPC with dexmedetomidine in experimental small intestinal ischaemia in horses.

General comments.

The study was well designed and adequately performed. The results were well described, and the statistical analysis was well executed. In the discussion, data were well-argued, and references were up to date.

In my humble opinion, the issue of I/R injury in colic surgery in the horse is still to be investigated, and this study adds a novel contribution to the knowledge of this topic. For this reason, the paper should be worth to be published as it.

6. PLOS authors have the option to publish the peer review history of their article (what does this mean?). If published, this will include your full peer review and any attached files.

Reviewer #1: Yes: Brigitte von Rechenberg, Prof. Dr.med.vet. Dipl. ECVS

Reviewer #2: No

---

## [Author Response · Author response to Decision Letter 0]

17 Mar 2020

As desribed in the Revision letter We tried to cover all required changes. Thanks again for taking the time to review our paper.

---

## [Decision Letter · Decision Letter 1]

9 Apr 2020

Ischaemic preconditioning and pharmacological preconditioning with dexmedetomidine in an equine model of small intestinal ischaemia-reperfusion

PONE-D-19-29159R1

Dear Dr. Sabine Kästner,

We are pleased to inform you that your manuscript has been judged scientifically suitable for publication and will be formally accepted for publication once it complies with all outstanding technical requirements.

With kind regards,

Francesco Staffieri

Academic Editor

PLOS ONE

Additional Editor Comments (optional):

Reviewers' comments:

Reviewer's Responses to Questions

**Comments to the Author**

1. If the authors have adequately addressed your comments raised in a previous round of review and you feel that this manuscript is now acceptable for publication, you may indicate that here to bypass the “Comments to the Author” section, enter your conflict of interest statement in the “Confidential to Editor” section, and submit your "Accept" recommendation.

Reviewer #1: All comments have been addressed

2. Is the manuscript technically sound, and do the data support the conclusions?

Reviewer #1: Yes

3. Has the statistical analysis been performed appropriately and rigorously? 

Reviewer #1: Yes

4. Have the authors made all data underlying the findings in their manuscript fully available?

Reviewer #1: Yes

5. Is the manuscript presented in an intelligible fashion and written in standard English?

Reviewer #1: Yes

6. Review Comments to the Author

Reviewer #1: All comments have been addressed to my satisfaction. Except the last comment in the acknowledgments has not been addressed. The question was, whether the foundation "ProPferd" should also be acknowledged, since the authors received also funding from there. Should be checked with the authors.

7. PLOS authors have the option to publish the peer review history of their article (what does this mean?). If published, this will include your full peer review and any attached files.

Reviewer #1: Yes: Brigitte von Rechenberg, Prof. Dr.med.vet., Dipl. ECVS

---

## [Editor Report · Acceptance letter]

13 Apr 2020

PONE-D-19-29159R1 

Ischaemic preconditioning and pharmacological preconditioning with dexmedetomidine in an equine model of small intestinal ischaemia-reperfusion 

Dear Dr. Kästner:

I am pleased to inform you that your manuscript has been deemed suitable for publication in PLOS ONE. Congratulations! Your manuscript is now with our production department. 

With kind regards,

on behalf of

Dr. Francesco Staffieri 

Academic Editor

PLOS ONE